# Optimal immune specificity at the intersection of host life history and parasite epidemiology

Alexander E. Downie [1]*, Andreas Mayer [2], C. Jessica E. Metcalf [1,3], Andrea L. Graham [1]

**1** Department of Ecology & Evolutionary Biology, Princeton University, Princeton, New Jersey, United States of America, **2** Lewis-Sigler Institute for Integrative Genomics, Princeton University, Princeton, New Jersey, United States of America, **3** School of Public and International Affairs, Princeton University, Princeton, New Jersey, United States of America

* adownie@princeton.edu

**Data Availability Statement:** The R code used in this paper is available as a supplemental file with this manuscript. The COMADRE database of animal life history matrices is available at https://

## Abstract

Hosts diverge widely in how, and how well, they defend themselves against infection and immunopathology. Why are hosts so heterogeneous? Both epidemiology and life history are commonly hypothesized to influence host immune strategy, but the relationship between immune strategy and each factor has commonly been investigated in isolation. Here, we show that interactions between life history and epidemiology are crucial for determining optimal immune specificity and sensitivity. We propose a demographically-structured population dynamics model, in which we explore sensitivity and specificity of immune responses when epidemiological risks vary with age. We find that variation in life history traits associated with both reproduction and longevity alters optimal immune strategies–but the magnitude and sometimes even direction of these effects depends on how epidemiological risks vary across life. An especially compelling example that explains previously-puzzling empirical observations is that depending on whether infection risk declines or rises at reproductive maturity, later reproductive maturity can select for either greater or lower immune specificity, potentially illustrating why studies of lifespan and immune variation across taxa have been inconclusive. Thus, the sign of selection on the life history-immune specificity relationship can be reversed in different epidemiological contexts. Drawing on published life history data from a variety of chordate taxa, we generate testable predictions for this facet of the optimal immune strategy. Our results shed light on the causes of the heterogeneity found in immune defenses both within and among species and the ultimate variability of the relationship between life history and immune specificity.

## Author summary

Organisms must use their immune defenses to counter infections, and their particular immune needs and optimal strategy for defense are likely to depend on the infection threats they face and their reproductive and survival schedules. Yet little is known about how these factors might interact together to influence immune strategy. We used a population dynamics model to examine how optimal immune specificity in host recognition of

compadre-db.org/Data/Comadre; we used Version 4.21.1.0 of the database.

**Funding:** A. E. D. acknowledges funding support from the US National Science Foundation Graduate Research Fellowship Program (Award Number DGE-2039656) (https://nsf.gov/). A. M. acknowledges funding support from a Princeton University Lewis-Sigler Fellowship (https://lsi. princeton.edu/). The funders had no role in study design, data collection and analysis, decision to publish, or preparation of the manuscript.

**Competing interests:** The authors have declared that no competing interests exist.

parasites depends simultaneously upon reproduction, survival, and parasite threats across life. We find that the strength and direction of the association between immune specificity and reproduction or survival depends on parasite threats and how they vary with age. For example, a highly specific immune response is favored for long-lived, slow-reproducing organisms when infection risk declines with age but is also favored for short-lived, fast-reproducing organisms when infection risk rises with age. Thus organisms with very different life history schedules may have identical specificities depending on their particular circumstances. Our research highlights how the immune strategies needed to "live fast and die young" or "live long and prosper" are not fixed but rather will depend on the interplay of different pathogen risks at different stages of life with when, during their lives, organisms reproduce.

## Introduction

Parasites are a central threat to organismal fitness, responsible for a considerable share of mortality. Accordingly, all organisms possess some anti-parasite capacities, which can be brought together under the umbrella of immune defenses. We expect immune defenses to be tailored to a host's ecological context to maximize efficacy and efficiency, particularly since immune defense incurs resource, immunopathology, and other costs [1–5]. Accordingly, the strategies hosts use may differ. For example, should parasites be directly resisted to extirpate them, or should the damage they inflict be tolerated and contained without directly countering the parasites [6,7]? Should immune investment be constitutive or inducible [4]? Given immunopathology, should parasite recognition systems be very sensitive to possible threats or highly conservative and specific in identifying them [8]? Different strategies imply deployment of different immune defenses and in different quantities, which will in turn have different impacts on parasites. In principle this applies to both inter- and intraspecific comparisons–there should be variation in immune strategies within populations and across species. Discovery of the causes of such variation will enable predictions for how organisms differ in their immune defenses and explain previously confounding empirical phenomena, as well as a better understanding of the vulnerabilities and strengths of any given immune defense strategy.

But what factors might shape immune strategy? One commonly-hypothesized ecological factor affecting immune strategy and deployment is epidemiology [9–13]. For example, we expect investment in immune defenses to rise with increasing parasite threat [10]–without parasites, why waste resources on immune defense? Ecological feedbacks between parasite dynamics and immune defenses can have intriguing non-linear effects on optimal immune strategies, including negative frequency-dependent selection on specific defenses [12,13]. Furthermore, we expect variability and diversity in parasite threats to affect the apportionment of that investment to different defenses [9,10,14] and the architecture of those defenses [15,16]. Immunopathology–damage inflicted on the self by the immune system–also presents a significant risk to hosts; balancing immunopathology with averting parasite-inflicted damage may constrain investment in immune defenses and alter their form [8,17–19]. While assessing details of epidemiological risk in empirical settings can be difficult and relatively few studies have attempted to look across multiple taxa, empirical work has generally supported the influence of disease environment on optimal immune defense and strategy [19–21].

Host life history has also been proposed to affect immune strategy [1,3,22–24]. This idea springs from the aforementioned resource costs of immunity, since life history is intimately tied to resource allocation [25]. Correspondingly, immune defenses have been proposed to be

linked to the fast/slow spectrum of life history and the related pace-of-life concept [22,23]. In particular, immune investment is predicted to correlate negatively with reproductive output [1,3,23] and positively with longevity [23]. Furthermore, Lee [23] suggests that species with a faster life history may use more specific immune defenses and fewer inflammatory immune defenses. A body of theory predicts ties between immune strategy and survival or reproduction, [13,18,26], but the life history traits are usually modeled as simple rates of reproduction and mortality per unit time that are constant across life, rather than variable as in most populations [27]. Experimental evidence supports a negative trade-off between immune investment and reproductive effort [28–30], but generally, empirical investigations of the relationship between immune strategy, the type of immune defenses used, and life history have been inconclusive [31–33]. Despite this, predictions of consistent relationships between pace-of-life and immune strategy remain common organizing hypotheses in disease ecology [24,33].

While epidemiology and life history are both considered important for immune defense, their intersection has rarely been explored. The few studies that exist have uncovered intriguing contingencies for the relationship between immunity, epidemiology, and life history. For example, Donnelly et al. [34] have shown that the diversity of parasites in a host's environment can shape the relationship of lifespan with immune memory. Even in one-parasite systems, optimal investment in memory peaks at intermediate lifespans due to ecological feedbacks between immune defenses and parasite transmission and prevalence, as well as, in some models, the contribution of herd immunity [12,26,35–37]. In two-parasite systems, that effect disappears, and optimal memory investment increases with increasing lifespans [34]. Thus, the epidemiological context (here the diversity of parasites) shapes the relationship between life history (lifespan) and optimal immune strategy (immune memory).

These results suggest that life history and epidemiology interact strongly to determine optimal immune strategies and may thereby explain why said strategies vary. Here we are especially interested in variation in epidemiological environment with age. Empirical work shows that variation in infection risk and burden with age is widespread, with different profiles of age-dependence for different host and parasite species [38–40]. For example, in Cape buffalo (*Syncerus caffer*), infections during the first year of life are largely restricted to tick-borne parasites [41]; in Seychelles warblers (*Acrocephalus sechellensis*) the prevalence of *Haemoproteus* blood parasites is greatest in juveniles [42]. And sexually transmitted infections will generally only pose a threat to hosts after reproductive maturity [43]. Immune strategies may be pulled in multiple directions by different infection threats at different stages of life. But studies of the effect of epidemiological context on immune defense generally neglect variation in disease risks with age. Therefore, by studying the intersection of age-dependent epidemiological variation with life history variation, we aim to elucidate causes of varied immune strategies across host taxa.

To this end we build on the model of immune sensitivity and specificity developed in Metcalf et al. [8] and Metcalf and Graham [44] by considering age-associated changes in infection risks and a broad range of life histories, particularly in reproductive demography, drawn from empirical research. The logic of the basic model is as follows. Hosts face a variety of antigenic stimuli, from harmful sources–generally non-self–and benign sources. The host must correctly ascertain from these stimuli whether or not it is infected and should mount an immune response (Fig 1). As in other signal detection systems [45,46], a trade-off between sensitivity and specificity then arises: an organism with a more sensitive immune system will more frequently correctly identify parasite stimuli and attack them, but it will also more often produce immunopathology by mistaking self molecules for parasite signals. An organism that is more specific is less likely to produce a damaging autoimmune response, but it is also more likely to ignore a parasite and therefore suffer more parasite-inflicted damage. Sensitivity and

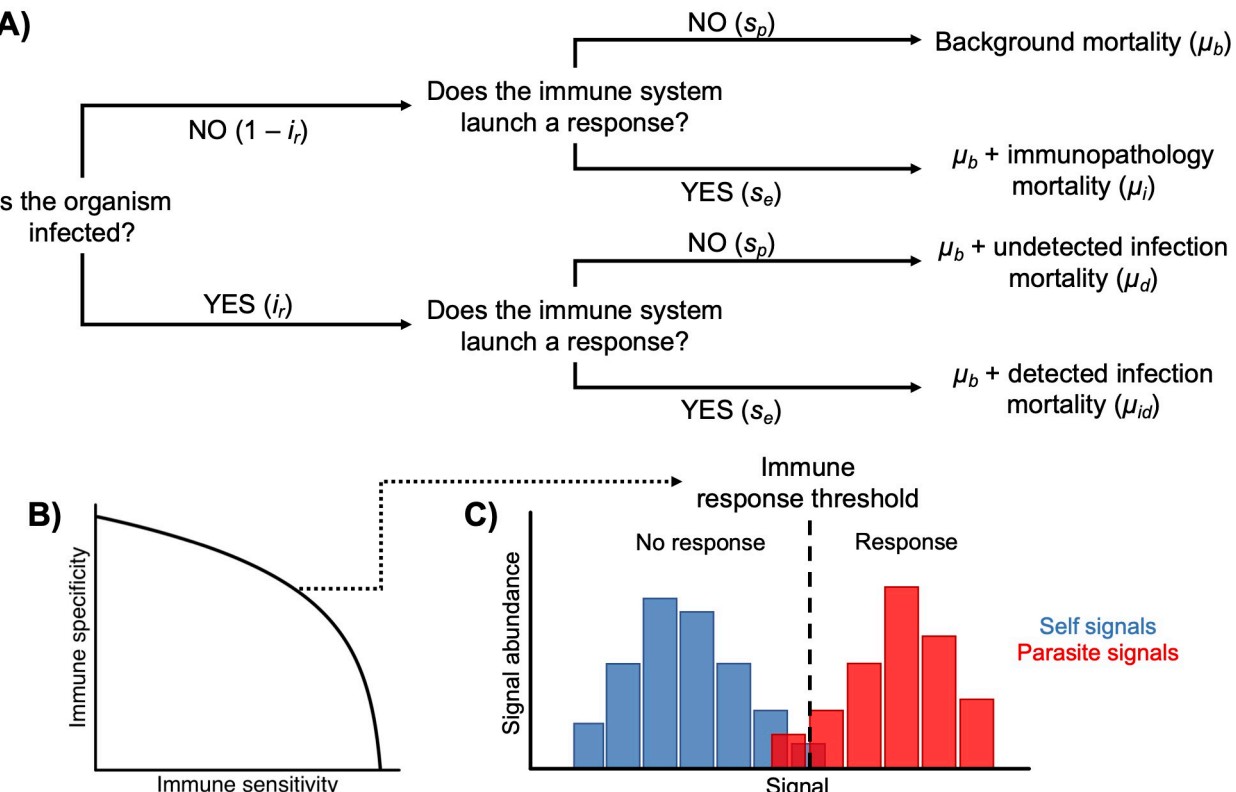

**Fig 1. Sensitivity and specificity in immunity.** A) Decision tree outlining the relationship of sensitivity and specificity to infection and mortality risks. Sensitivity and specificity, together with frequency of infection, determine the relative balances of various types of mortality. B) Receiver-operator curve showing trade-off between sensitivity and specificity in immunity. Different points on trade-off curve indicate different combinations of sensitivity and specificity and produce different response thresholds. The shape of the trade-off curve is governed by how much host and parasite signals overlap and the capacity of the host to discern fine molecular differences. C) Schematic showing how sensitivity and specificity manifest in response to distributions of benign (blue) and pathogenic (red) signals. Different combinations of sensitivity and specificity define the location of the response threshold. A more sensitive immune system has a response threshold shifted to the left, responding to a greater range of signals; a more specific immune system has a response threshold shifted to the right, responding to a smaller range of signals.

specificity may be encoded in receptor structure or in activation thresholds for the immune system [8,19] or possibly even in the range of defenses deployed against different parasites [47]. Although relatively little research has directly explored immune defenses from this perspective, such trade-offs have been documented: for example, greater reactivity to non-self-antigens (i.e. sensitivity) correlates with greater reactivity to self antigens [48], suggesting a trade-off between susceptibility to infection and susceptibility to immunopathology. Cross-reactivity (i.e., lack of specificity) of T cell, B cell, and antibody responses contributes to the immunopathology underlying severe dengue hemorrhagic fever [49] and might be a causal explanation for why infection sometimes brings about autoimmune disease [50].

Combining sensitivity and specificity with epidemiological risks–infection risk, infection mortality risk, and immunopathology risk–allows us to determine survival in a given risk environment (Fig 1A). We can then use a demographic matrix framework to provide resolution on both life history and epidemiological variation. Demographic matrices–sometimes called population projection matrices or Leslie matrices–are a general tool for framing and quantifying population outcomes (population growth, stage, or age structure) when demographic processes (survival, reproduction) vary by age or stage [51]. The optimal immune strategy is that which maximizes fitness; we define fitness here as the population growth rate, λ, which

describes the population rate of increase through time and is the dominant eigenvalue of the population projection matrix. As we initially consider static epidemiological contexts, and include no explicit forms of density dependence among hosts within our matrix models, the success of any host immune strategy will not be contingent on strategies being used by other individuals in the population. This framing makes a focus on optimizing λ appropriate, rather than developing resistance-to-invasion analyses to identify Evolutionarily Stable Strategies (ESS) [9–13].

Prior work has shown that the relative balance of the different types of epidemiological threats shapes immune strategy: in general, an organism facing greater infection risk should be more sensitive, and an organism facing lower infection risk should be more specific [8,44]. Unlike some models exploring other aspects of immune strategy (e.g. [13]), our framework does not currently include ecological feedbacks on disease transmission, though incorporating such feedbacks would be a fascinating vein for future work. Our simpler epidemiological risk parameters could be held to represent a broad suite of pathogens posing some overall risk to the host of infection-induced mortality. Furthermore, in our framework, background mortality does not affect sensitivity and specificity, except when disease or immunopathology risks vary with age [8,44]. When such risks do not vary with age, then no aspect of demography affects immune specificity and sensitivity [44], although background mortality can, in other modeling frameworks, affect other aspects of optimal immune strategy [34, 36]. However, as discussed above, disease and immunopathology risks are rarely static.

Here we explicitly study the interaction between life history and epidemiology as it influences optimal immune strategy, drawing on a variety of scenarios reflecting empirical patterns. We mix variation in life history strategies with variation in epidemiological risk across life to bring potentially-influential nuance to both life history and disease risk within life. First we focus on different simple reproductive output schedules, exploring how they shape optimal immune specificity and sensitivity in various epidemiological contexts. We then marry a range of realized life histories drawn from the COMADRE database of demographic matrices [52] with epidemiological risk schedules to explore in greater detail the complex interplay between life history, epidemiology, and immune strategy. We find that this confluence is indeed critical, with epidemiology mediating both the strength and direction of the relationship between life history traits and optimal immune strategy.

## Results

### Reproductive demography affects optimal immune strategy

We explored five different schedules for reproduction in our analysis of the relationship between reproductive demography and optimal immune strategy (Fig 2). Each reproductive schedule gives a different set of fertility parameters for the age classes in a demographic matrix. For a given set of epidemiological risk parameters, each reproductive schedule has a different associated optimal immune sensitivity and specificity maximizing the population growth rate (λ) (Fig 2). In the case considered here, where infection risk drops at the age of first reproduction, a life history associated with higher reproductive output leads to lower sensitivity and higher specificity, and vice-versa for lower reproductive output. The two reproductive schedules having changes in output with age are further polarized in their associated optimal immune strategies (Fig 2). Each reproductive schedule and optimal immune strategy also associates with different stable age structures and distributions of reproductive value and the elasticities of λ to both survival and fecundity (S1 Fig). These elasticities are defined as the proportional change in λ produced by a change in survival or fecundity parameters, respectively [51]. Because different reproductive schedules produce different values of λ, we also

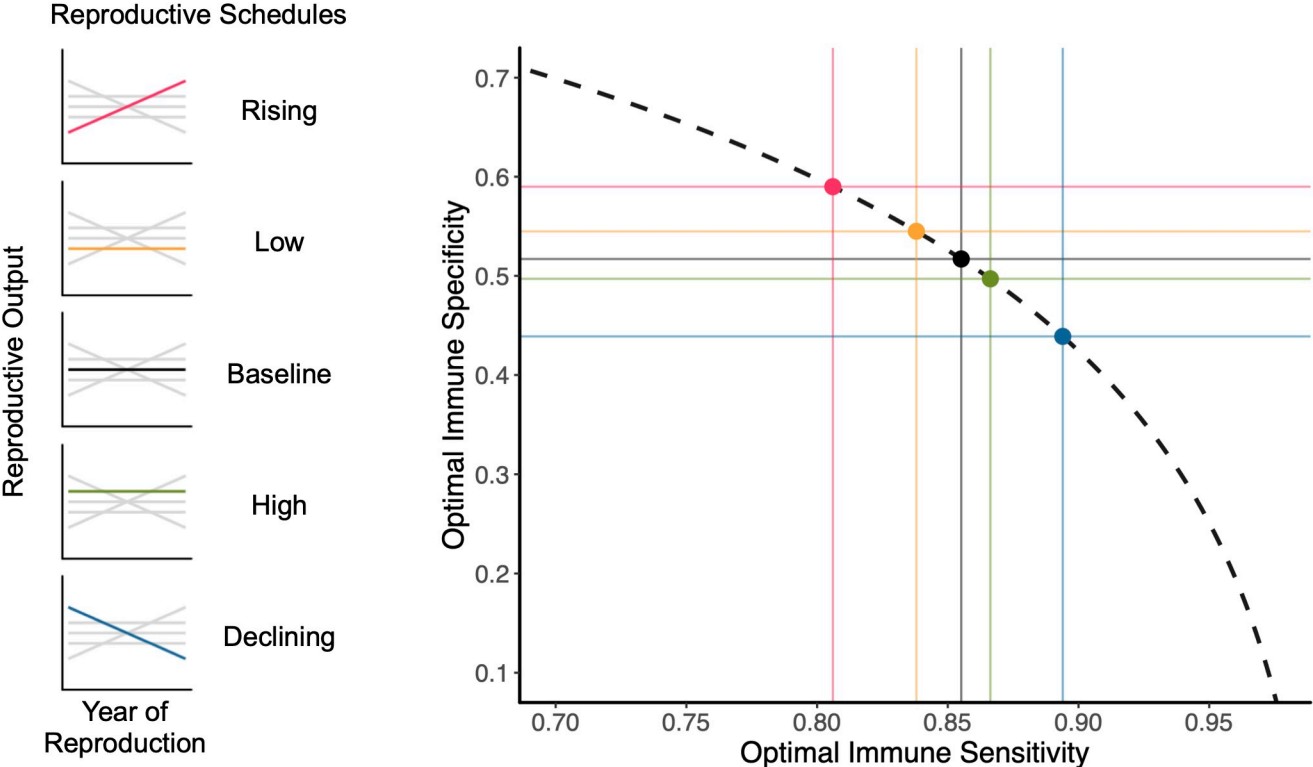

**Fig 2. Influence of reproductive demography on optimal immune strategy.** Plot showing optimal combination of immune sensitivity and specificity for each of five reproductive demographic schedules (color-coded at left), for an epidemiological risk environment where infection risk $i_r = 0.6$ for the first two age classes (prior to reproductive maturity) and $i_r = 0.2$ thereafter. Reproduction begins in the third age class for all demographic schedules. Strategy optima, shown as points on the dashed curve, are determined as the immune specificity and sensitivity maximizing $\lambda$, the population growth rate. Dashed curve shows the shape of the specificity/sensitivity trade-off curve for $\gamma = 4$. Solid lines show values of the respective optimal strategies on each axis. Other parameter values are $\mu_b = 0.15$, $\mu_i = 0.1$, $\mu_d = 0.3$, and $\mu_{id} = 0.01$.

explored optimal immune strategies when we manipulate background mortality ($\mu_b$) to equalize $\lambda$ across the reproductive schedules; the effect of reproductive schedule on immune strategy is unaltered (S1 Table). Thus, it is not $\lambda$ that is associated with a given immune strategy, it is the reproductive schedule itself.

## Epidemiological risk environment modulates effects of reproduction on immune strategy

Different organisms experience different epidemiological risks across their lives. We therefore expanded on the analysis above by exploring how the relationship between reproductive demography and immune strategy changes in different epidemiological scenarios (Fig 3). We find that a different epidemiological context can completely reverse the direction of this relationship (Fig 3A). In an epidemiological setting with infection risk ($i_r$) decreasing at reproductive maturity from a high level to a low level, the late-skewed rising reproductive schedule (red) has the most specific immune strategy of our five reproductive schedules, whereas when $i_r$ rises at reproductive maturity from low to high, that same reproductive schedule associates with the least specific immune strategy. The early-skewed declining reproductive schedule (blue) has the opposite set of associations in these two epidemiological environments. When infection risk is high in early life and low in late life, the rising schedule favors greater immune specificity than the declining schedule, and vice-versa. In essence, a "flip" in the ordinal

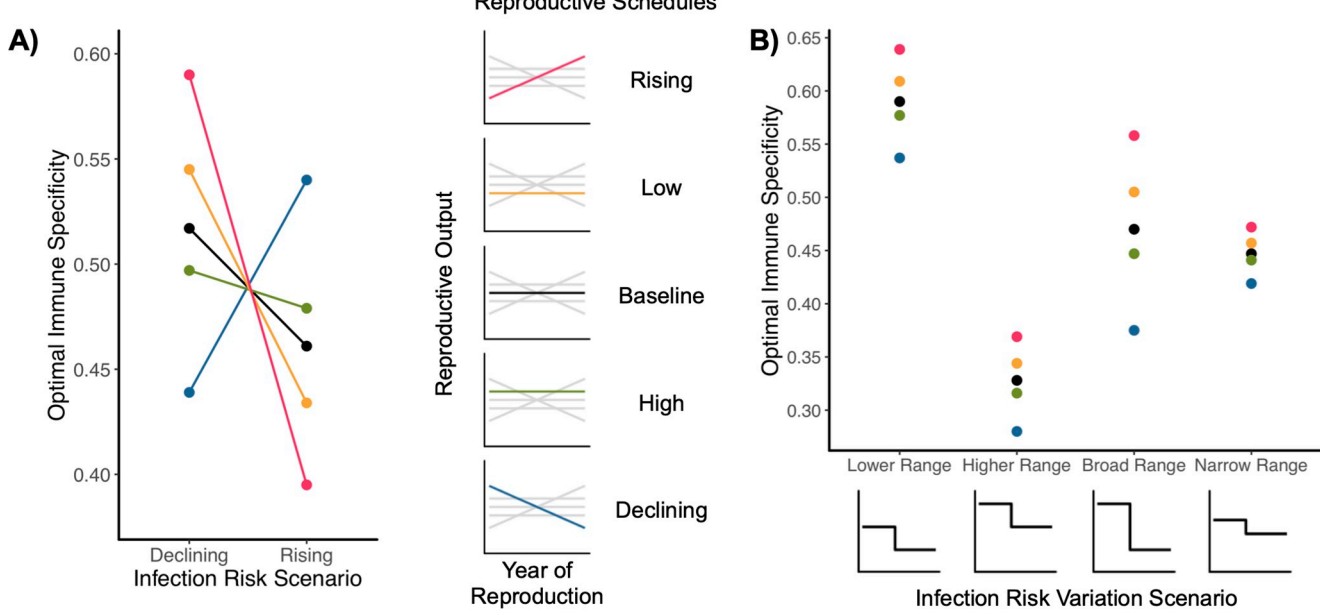

**Fig 3. Epidemiological environment alters the effect of reproduction on immune strategy: infection risk $i_r$.** Reproduction begins in the third age class for all schedules. In each scenario, $i_r$ starts at one value and changes to a different value at the third age class. A) The change in optimal immune specificity associated with differences in epidemiological context (i.e. changes in $i_r$, on the x-axis) and reproduction (different points and lines, color-coded at center). In the declining scenario, $i_r$ drops at reproductive maturity from 0.6 to 0.2; in the rising scenario, $i_r$ increases from 0.2 to 0.6. Other parameter values are $\mu_b$ = 0.15, $\mu_i$ = 0.1, $\mu_d$ = 0.3, $\mu_{id}$ = 0.01, and $\gamma$ = 4. B) The change in range of optimal specificities associated with different reproductive demographies for different magnitudes of variation in decline of infection risk $i_r$ at reproductive maturity. In the lower range scenario, $i_r$ drops from 0.45 to 0.2; in the higher range, from 0.7 to 0.45; in the broad range, from 0.7 to 0.2; in the narrow range, from 0.525 to 0.375. Other parameter values are $\mu_b$ = 0.15, $\mu_i$ = 0.1, $\mu_d$ = 0.3, $\mu_{id}$ = 0.01, and $\gamma$ = 4.

relationship between reproductive and immune strategies takes place when organisms experience the same risks but at different points in life–i.e. when the epidemiological schedule is reversed. This flipping effect also appears when we explore variation in undetected infection mortality risk ($\mu_d$) across life while holding $i_r$ constant (S2A Fig).

Differences in the range and magnitude of epidemiological risk variation with age alter the differences in optimal immune strategy between reproductive demographies. Overall level of risk produces different immune strategies, and when the magnitude of variation across life changes, the strength of the reproduction-immunity relationship changes (Fig 3B). Here, greater $i_r$ variation across life is associated with greater differences in immune strategy between reproductive schedules. Variation in other risk parameters produces the same insight: when we examined variation in $\mu_d$, we found the same correlation between the strength of the reproduction-immune strategy relationship and degree of epidemiological risk variation (S2B Fig). Overall, our results suggest we cannot know from life history alone which of two populations or species should be more sensitive or specific in immune defense without knowing the epidemiological context of each population.

## Epidemiology interacts with several demographic traits to influence immune strategy

To expand on our above results, we considered the relationship between specific life history traits and optimal immune strategy. We did this by exploiting the life histories recorded in the COMADRE database of animal demographic matrices [52,53]. These matrices provide reproductive schedules and estimated mortality curves across a wide taxonomic range, and we

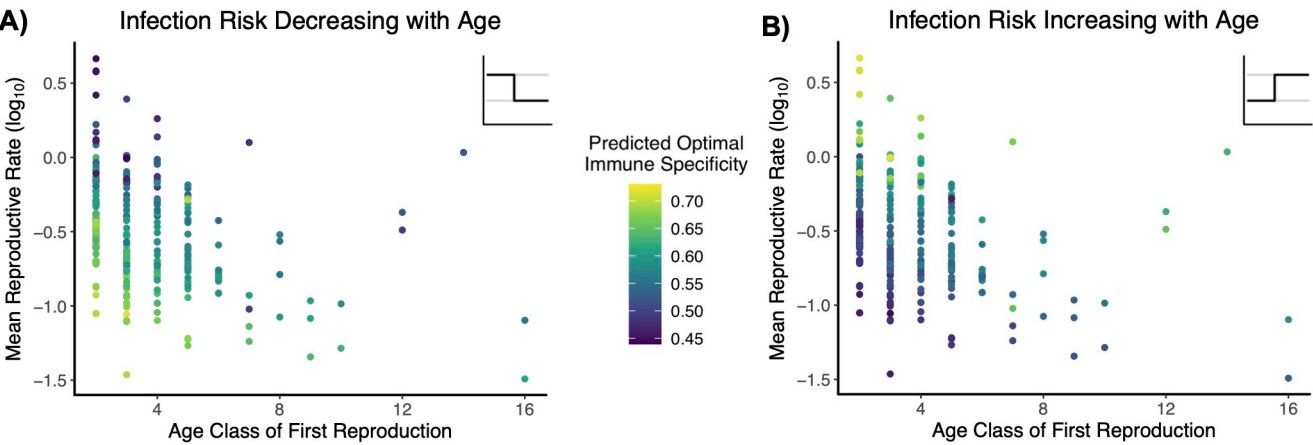

**Fig 4. Interaction of demography and epidemiology for immune specificity.** Infection risk is set such that there is one infection risk $i_r$ for age classes prior to reproductive maturity and a different risk $i_r$ for reproductive age classes. Our dataset comprises 298 population matrices representing 129 chordate species. For all scenarios, parameter values are $\mu_d = 0.3$, $\mu_i = 0.1$, $\mu_{id} = 0.01$, $\rho = 0.75$, and $\gamma = 4$. A) Predicted optimal immune specificities when infection risk declines with age, with respect to population age class of first reproduction and mean reproductive rate as calculated from original matrix. Infection risk $i_r$ prior to reproductive maturity is 0.45; for reproductive age classes, it is 0.2. B) Predicted optimal immune specificities when infection risk rises with age, with respect to population age class of first reproduction and mean reproductive rate as calculated from original matrix. Infection risk $i_r$ prior to reproductive maturity is 0.2; for reproductive age classes, it is 0.45.

processed the raw matrices to mesh them with our immune strategy estimation approach without changing the essential demographic patterns encoded in said matrices (see Methods, S3 Fig). Each age class in the demographic matrices corresponds to a year of life, rather than a life stage of some variable length of time. We combined the output matrices with various epidemiological scenarios to explore, in a given epidemiological context, the change in predicted optimal immune strategy among different demographies. We analyzed the relationship between predicted optimal immune specificity and life history traits with a Bayesian linear model. For epidemiological variation we used stepped variation in $i_r$ as above, where $i_r$ changes once at reproductive maturity. We used fixed values for the other parameters in our method; while there is no *a priori* reason to expect parasite mortality threats and immune system functioning to be identical across a wide range of taxa, this enables a mathematically-controlled comparison. In preliminary work we explored other combinations of parameter values and did not find any differences between sets of parameter values that affect our qualitative results.

We find correlations between predicted optimal immune strategy and three different life history traits–age class of first reproduction, mean reproductive rate, and reproductive life expectancy (Figs 4 and 5 and Table 1). And we identify the same epidemiology-induced flip in the life history-immune strategy relationship that we describe above. The effects of the life history traits are interlocking, such that variation unexplained by one life history trait may be explained by another. An early age class of first reproduction can be associated with a wide range of optimal immune strategies, but these strategies are in turn shaped by the mean reproductive rate (Fig 4). For decreasing infection risk with age and an early age class of first reproduction, high specificity corresponds to low rates of reproduction, and vice-versa. At later age classes of first reproduction, the range of associated reproductive rates contracts, and optimal specificity becomes less variable and confined to relatively higher values (Fig 4A). The reverse pattern is observed when infection increases with age (Fig 4B and Table 1).

In general, the strength and direction of the relationship between any given life history parameter and predicted optimal immune strategy depends on the level of infection risk and the amount it varies with age (Fig 5). When risk does not vary at all with age, then there is no

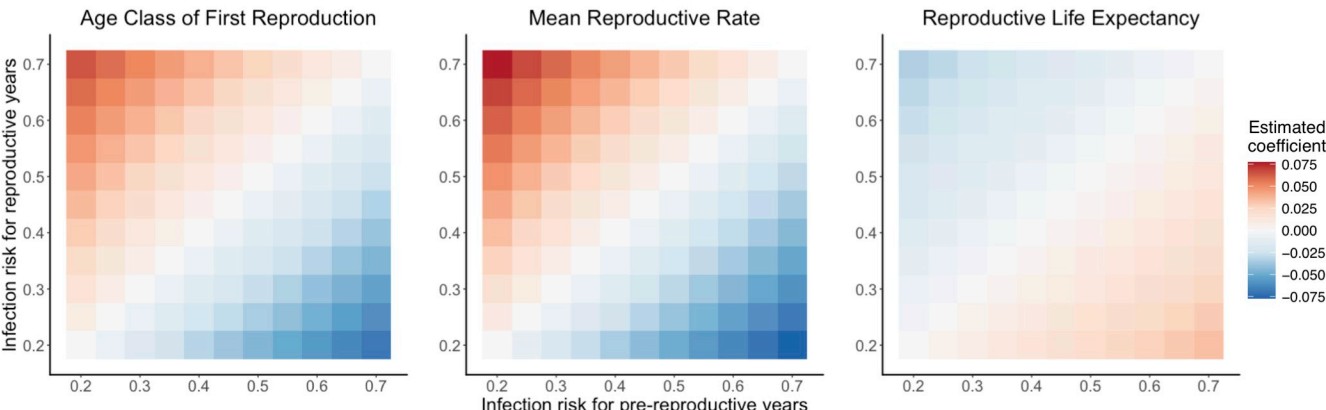

**Fig 5. Epidemiological dependency of life history-immune specificity relationship.** Tile plots showing, for a variety of scenarios of variation in infection risk $i_r$, coefficients of relationship between the designated life history trait and predicted optimal immune specificity as estimated from Bayesian linear models. Each tile is a scenario in which predicted optimal immune specificities were generated for 298 population projection matrices representing 129 chordate species. A linear model was used to estimate relationship coefficients for each scenario. The coefficient value in the plot represents the mean value of the posterior probability distribution, except on the diagonal; on the diagonal, all coefficients are 0 (see text). The value of $i_r$ on the x-axis is the risk for pre-reproductive years, jumping in age class of first reproduction to the value of $i_r$ on the y-axis. Life history trait values were log-transformed and standardized as Z-scores for comparability of coefficients. For all estimated coefficients off the diagonal, 89% credible intervals do not include 0. For all scenarios, parameter values are $\mu_d = 0.3$, $\mu_i = 0.1$, $\mu_{id} = 0.01$, $\rho = 0.75$, and $\gamma = 4$.

relationship between any individual demographic trait and immune specificity, regardless of absolute risk level (as previously described). However, as the magnitude of change in infection risk at reproductive maturity grows, the strength of each immunity-demographic trait relationship increases, with the sign of the relationship depending on which trait is being examined and whether risk increases or decreases at reproductive maturity. The estimated relationship between mean reproductive rate and immune specificity is the strongest in all infection risk scenarios, followed by age class of first reproduction and then reproductive life expectancy (Fig 5 and Table 1).

We further examined the robustness of our central finding with different datasets and other life history traits. The epidemiological dependency of the life history-immune strategy

**Table 1. Results from Bayesian linear model for demography and immune specificity.** Linear model looks at predicted optimal immune specificity as a function of three different life history summary statistics. Results are means and, in brackets, boundaries of 89% highest posterior density intervals (HPDI) for posterior probability distributions for parameter values. Entries in *italics* indicate the 89% HPDI overlaps with 0 for that parameter. All summary statistics were calculated from original matrix in COMADRE database, log-transformed, and standardized as Z-scores. Dataset includes 298 qualifying matrices from 129 chordate species. For stepped epidemiological scenario, when infection risk is rising, $i_r$ in pre-reproductive years is 0.2, and $i_r$ in reproductive years is 0.45. When infection risk declines in the stepped scenario, $i_r$ in pre-reproductive years is 0.45, and $i_r$ in reproductive years is 0.2. In smoothed declining scenario, $i_r$ declines from 0.45 to 0.2; in rising scenario, $i_r$ rises from 0.2 to 0.45. Other parameter values are $\mu_d = 0.3$, $\mu_i = 0.1$, $\mu_{id} = 0.01$, $\rho = 0.75$, and $\gamma = 4$.

| Parameter | Declining stepped infection risk $i_r$ | Rising stepped infection risk $i_r$ | Declining smoothed infection risk $i_r$ | Rising smoothed infection risk $i_r$ |
|---|---|---|---|---|
| **Intercept** | 0.603 [0.600, 0.605] | 0.544 [0.541, 0.547] | 0.538 [0.536, 0.541] | 0.608 [0.605, 0.611] |
| **Age class of first reproduction** | -0.0381 [-0.0411, -0.0352] | 0.0357 [0.0327, 0.0384] | *4.82x10^{-4}* *[-0.00285, 0.00380]* | *-0.00121* *[-0.00463, 0.00239]* |
| **Mean reproductive rate** | -0.0423 [-0.0454, -0.0390] | 0.0412 [0.0379, 0.0444] | -0.0409 [-0.0444, -0.0374] | 0.0440 [0.0402, 0.0479] |
| **Reproductive life expectancy** | 0.0190 [0.0155, 0.0224] | -0.0184 [-0.0218, -0.0150] | -0.0232 [-0.0269, -0.0197] | 0.0252 [0.0211, 0.0292] |
| **Standard deviation** | 0.0278 [0.0260, 0.0298] | 0.0280 [0.0260, 0.0300] | 0.0304 [0.0285, 0.0324] | 0.0322 [0.0301, 0.0344] |

relationship is recapitulated when we repeat our analysis with a dataset of only mammal life histories (S2 Table) and when we consider generation time as the sole life history trait influencing immune strategy in our linear model (S3 Table). When we calculate our life history statistics from our matrices after dimension manipulation and mortality curve approximation (see Methods), results are qualitatively the same while differing to a small degree quantitatively (S4 Table). This highlights that our matrix processing method, while producing matrices that approximate the original COMADRE matrices but are not identical, is not inducing significant distortions and generating mirages. All these analyses reinforce our main result, that epidemiological context determines the strength and direction of the relationship between life history traits and optimal immune strategy.

### The interaction between epidemiology and life history for immune specificity spans multiple modes of risk variation

We also explored an alternate mode of infection risk variation with age. In this mode, rather than infection risk changing just once at reproductive maturity, we defined a risk for the first age class of the matrix and a risk for the last age class and allowed risk to change at a constant rate across intermediate age classes (see Methods, S4 Fig). This scenario represents a smooth change of infection risk with age. For use with our COMADRE process, we adjust the slope such that the relative change in infection risk with respect to matrix dimension (itself determined as a function of age at first reproduction and reproductive life expectancy) stays constant, ensuring comparability between matrices with different life histories (S4 Fig).

We repeated all of the preceding analyses for this second mode of infection risk variation, and we identified the same major results. Reproductive demography does influence optimal immune specificity and sensitivity (S5 Fig and S5 Table), with the strength and direction of that influence depending on the direction and magnitude of change in epidemiological risk parameters (S6 and S7 Figs). Furthermore, when we extend to our broader demographic analysis, we find the same epidemiological dependency for the strength and direction of the relationship between immune specificity and specific life history traits (S8 and S9 Figs). Intriguingly, we do find different relative and absolute magnitudes for each relationship when compared to the stepped mode of infection risk variation (S9 Fig and Tables 1 and S2–S5). If infection risk varies in the smoothed manner, our linear models are not confident that age class of first reproduction has an influence on optimal immune strategy (Tables 1 and S2–S5). And the directions of the relationships between optimal immune specificity and reproductive life expectancy are reversed (S9 Fig). This further highlights our finding that how epidemiological context varies with age shapes the relationship between life history traits and immune strategy. Particular demographic traits may have more or less influence on immune strategy, and in different directions, depending on precisely the epidemiological context.

### Discussion

Here, using a demographic model, we demonstrate that the influence of life history on immune specificity and sensitivity depends on the epidemiological context and the way that epidemiological risk varies across life. Both the strength and direction of the relationship, for a variety of different life history traits, are changeable. Thus, for example, when infection risk declines with age, a reproductive schedule with high output will push populations towards more sensitive and less specific immune strategies, while when infection risk rises with age, high reproductive output will select for less sensitive, more specific immune strategies. In environments where infection risk variation with age is high, life history is very influential; when such variation is low, it is less important. Infection risk variation could be created by several

factors–for example, sociality [54]. In non-social species we might expect infection risk to be relatively low pre-reproduction, with a rise at reproductive maturity associated with the increased contact of mating and sexually-transmitted infections. In such species, those with greater reproductive output should have less sensitive, more specific immune strategies than those with lower reproductive output. But closely-related social species might not have the same variation in infection risk across age; perhaps they have a higher frequency of early-life infections relative to infection in adulthood (see [54] for more discussion of sociality and infection). In these species, we would find that high reproductive output selects on immune strategy in the opposite direction, pushing for more sensitive, less specific defenses. We find a similar context-dependency for the relationship of immune sensitivity and specificity to age at first reproduction, a development- and longevity-associated trait, and reproductive lifespan, a longevity-associated trait. These results crystallize the importance of epidemiology for optimal immune strategy, contrary to the common assumption that particular life history traits are consistently associated with particular immune phenotypes regardless of epidemiological context [23,24,33].

## Nuances of life history and epidemiology are crucial for optimal immune strategy

By employing a broad variety of life histories, we study the forces shaping immune strategy in greater detail than previously explored. Most prior studies of immune defense have used a constant background mortality to define lifespan. Such studies have produced intriguing results [8,34,37], but two species with the same average or maximum lifespan may have radically different survival curves and therefore demographic patterns [55]. By here describing lifespan in terms of age of first reproduction and reproductive life expectancy, we have identified how different facets of lifespan may in fact select for different strategies of immune specificity. For example, our model tells us that two species with similar average lifespans but different ages at first reproduction and reproductive life expectancies should differ in the specificity of their defenses against infection. Our analysis may also prove illuminating for the study of sex differences in immune defense [3,44]; males and females within a species may differ demographically–and in infection risks–in such a way as to push them towards different immune strategies. In general, our results show that greater detail in life history and epidemiology offers greater insight into optimal immune defense.

One particularly intriguing result is that differences in reproductive schedule alone can influence optimal immune specificity when epidemiological risk varies with age. This surprising finding is explained by the relationship between the reproductive schedule and the relative contributions to fitness of different life stages. In our model, greater reproductive output leads to population growth rate ($\lambda$) being more heavily dependent on both survival and fertility in early age classes relative to later age classes (S1C and S1D Fig, [51]). Donnelly et al. [13] demonstrate in their model that optimal immune resistance should be tuned to favor stages that contribute the most to the rate of population growth. Our model is not amenable to an analytical determination and examines a different aspect of immune strategy, but by analogy with Donnelly et al. [13], we might expect an immune strategy that boosts survival in those early stages to be optimal when reproductive output is higher, at the cost of survival in later stages. And, indeed, that is what we find: when infection risk declines with age, greater reproduction associates with greater sensitivity to mitigate greater infection risk in early life, when the elasticity of $\lambda$ to survival is highest ($e_x = \frac{\partial \ln(\lambda)}{\partial \ln(a_x)}$, where $a_x$ is the survival or fertility parameter of interest in age class $x$) [51]. The rising and declining reproductive schedules produce the most divergent elasticity distributions, weighted towards late and early age classes respectively, and

accordingly have the most divergent immune specificities. Just as the elasticity of population growth rate ($\lambda$) to survival may be informative for optimal immune specificity against mortality-causing parasites, so then might elasticity of $\lambda$ to fertility shed light on how specificity would be affected by parasites that depress fertility. Given such parasites, specificity should be similarly calibrated to mitigate threats in stages with the greatest elasticity to fertility, minimizing costs to fertility in those stages.

Thus reproduction may influence some aspects of optimal immune strategies through its effects on demographic structure in a population, independent of the resource allocation trade-offs often explored in models of immune strategy [18,28]. And because natural selection from threats across all life tunes immune strategy, the variation in epidemiological risk and host demography may produce apparent mismatches between immune strategy and the threats faced during given life stages, such that different mortality risks are more or less prominent at different times of life. This could be one source of immunosenescence, perhaps moderated by flexibility in sensitivity and specificity across age (see [8,44] for a more thorough discussion of the consequences of variation in immune sensitivity with age and immunosenescence).

Our results also suggest that theories predicting immune strategy primarily or only from pace-of-life or the fast-slow spectrum of life history [22–24] may not be accurate, at least for immune sensitivity and specificity. There are two reasons for this. First, the direction of the association between any given life history trait and optimal specificity is contingent upon epidemiology, and thus reproduction or longevity may not necessarily always have the same association with immune strategy or even, when infection risk does not vary across life, any association at all. Second, in certain epidemiological contexts, increased reproduction and increased longevity (which occupy opposite ends of the fast/slow spectrum) push immune specificity in the same direction. And age class of first reproduction and reproductive life expectancy have opposite associations with immune specificity in all epidemiological contexts we explored, despite both being associated with a slow pace of life [27]. Further work is necessary to establish whether this conclusion applies to other aspects of immune strategy, like resistance and tolerance. In particular, the aforementioned resource allocation trade-offs may influence different aspects of the immune system to greater or lesser degrees. Furthermore, particular life histories and ecological strategies may be associated with particular epidemiological risks [33,56], which may more closely tie life history to immune strategy. These are promising directions for future research.

Our incorporation of changes in infection risk with age allows fresh insights, but there are additional aspects of epidemiology that may affect optimal immune specificity and sensitivity. For example, co-evolution of host immune systems with individual parasite species can shape at least some aspects of host immune defense [e.g. 57]. In our framework, we neglect this process, favoring an approach that aggregates infection and mortality risk across a broad and static community of parasites. Different parasites may impose different selective pressures that wash out, making our approach a reasonable approximation of empirical selective pressures. However, it is also possible that co-evolution may be more influential for the specificity of immune memory–e.g., Schnaack and Nourmohammad [58] show that the extent of parasite evolution during an organism's lifespan shapes optimal specificity of memory against that parasite, which will trade off with the cross-reactivity of that memory against evolved variants of that parasite. Long-lived organisms with many encounters of a particular evolving pathogen should thus have more specific immune memory of that pathogen, because across long durations cross-reactivity becomes less valuable [58]. Indeed, repeated exposure may be an additional selective pressure that modulates evolved specificity in adaptive immune systems, in addition to the pressures that we describe here, although the precise concepts labeled "specificity" here

and in [58] are not exactly the same. A further selection pressure might be imposed by pathogens that occur irregularly with respect to age (including epidemic pathogens). Such pathogens would create fluctuating selection and may lead to a certain amount of standing variation in immune sensitivity and specificity within a population, with different phenotypes rising and falling in abundance as pathogen pressure ebbs and flows [59].

## Insights and opportunities for empirical ecoimmunology

Our findings, while focused on the one axis of immune strategy (specificity vs. sensitivity), may help explain why empirical investigations have struggled to consistently identify relationships between life history and immune defense when looking across multiple taxa. For example, lifespan correlates positively with immune investment in some [60] but not all studies [31,61,62] looking at various different suites of taxa. Our results here highlight two potential explanations for such variation. First, as described above, different aspects of lifespan produce different selective pressures on immune defense; thus the precise life histories of the species being compared are important context. As noted above, for cases where two organisms have similar mean lifespans but different ages at first reproduction and reproductive life expectancies, we would expect different optimal immune specificities. Second, in general these studies do not incorporate variation in epidemiological risk across taxa. Yet the disease ecology of each species shapes strategy to a great extent, especially via level of risk [8,10,13,21]. As long as epidemiological context varies across species–for example, infection risk rising with age in some species and declining with age in others–immune strategy may not correlate with life history per se. None of this is to say that life history does not matter–we clearly find that it can matter a great deal. But researchers studying variation in immune strategy should consider and examine both epidemiology and demography, including variation in infectious disease risk with age. Epidemiology is at present a hidden variable in many ecoimmunological studies; we should bring it to the light [63].

Empirical studies testing our predictions face several challenges but also present opportunities for advancing the fields of disease ecology and ecoimmunology. One such challenge is characterizing epidemiological risk and how it varies with age. Longitudinal studies like [41] and [42], though logistically challenging, may be our most promising avenue forward. Tractable force-of-infection models [64,65] may be particularly useful for establishing infection risk per unit time, which is highly influential in our analysis as it describes the frequency with which the immune system faces a challenge. An additional difficulty here is that the variation *within* a population in epidemiological risk is important for assessing the amount of resolution we can obtain in comparative studies. The greater the variation among individuals in epidemiological risk, the greater their predicted variation in immune strategy, which in turn would spill over to reduced differences among populations in the distributions of optimal immune specificities each contains.

Another challenge for empirically testing our results is the development of methods to characterize immune sensitivity and/or specificity. One potential avenue entails dose-response curves of immune defenses produced in response to antigen exposure or experimental infection [66,67]–these might offer a window into specificity and sensitivity as they would describe the threshold of receptor stimulation producing an immune response. Particular gene expression profiles, and how much they overlap across different antigenic stimuli, as discussed by Hawash et al. [47], offer another intriguing new approach. Ascertaining incidence of immunopathology may also provide insight into immune sensitivity, because ultimately immunopathology should be more common in more immunologically-sensitive but less pathogen-specific organisms. Alternatively, to understand receptor specificity and the range of antigens

provoking an immune response, one could directly examine binding affinities for toll-like receptors (TLRs) or the specificity of B and T cell receptors and the activation thresholds for cells thus bound [8]. Any of these approaches would help us identify sensitivity and specificity in identifying and responding to infections. Indeed, our results may even provide a framework to interpret empirical results already in the literature, such as delineating epidemiological scenarios underlying apparently greater immune sensitivity for apes as compared to monkeys [47].

### Broader applications in disease ecology

Our insights may also have value for researchers interested in global health and conservation of endangered taxa. The identification of reservoirs for zoonotic infections is a key research concern, one that has only been highlighted further by the COVID-19 pandemic [68–72]. To the extent that heterogeneities in zoonotic capacity exist, our model might help us to understand why, by highlighting the importance of demographic and epidemiological nuance in determining immune specificity and sensitivity, such that we can identify whether species housing a greater proportion of zoonoses have particular immune strategies as a consequence of their demography and epidemiology. For example, fast pace-of-life is hypothesized to be an indicator of zoonotic reservoir capacity, but evidence for this is mixed [33]; our results highlight how demographic and epidemiological differences within pace-of-life may create immune heterogeneities that could shape which species with a fast pace-of-life genuinely are important zoonotic reservoirs. Furthermore, the immune strategy of a zoonosis's reservoir host may be important for the level of threat that pathogen poses to humans [73–75]. For example, if indeed human immune systems are relatively sensitive, as suggested by Hawash et al. [47], then species with more specific immune systems may harbor zoonoses more dangerous to humans due to the mismatch between human and reservoir immune strategies. Our work offers a route to identifying these mismatches by a careful consideration of the comparative demography and epidemiology of species.

Lastly, while we use a demographic framework based on single-year age classes, and while we only focus on chordates, our modeling framework is taxon-agnostic. It is therefore quite plausible that our results apply more broadly, to many different organisms with lifespans on many different temporal scales. For example, do tree seedlings in the undergrowth experience different risks of infection and mortality than adults stretching into the canopy? How do the temporal durations of these different growth stages affect the weighting of such different hazards in shaping plant immune sensitivity and specificity? Considering these two factors might help us gain insight into the immune defenses deployed by plants and other organisms. Our results might also offer insight into the immune strategies of microbial organisms. Past research has described how epidemiology can shape prokaryote immune defenses [19,21,76], and our results may help us understand how risk variation and host strategy would contribute. The general principle that we find here is that immune specificity and sensitivity in chordates should be shaped according to how disease and immunopathology risks differ across ages and the contributions of those ages to fitness, and there is no reason that this would not apply across a wider range of host taxa.

### Conclusion

We have quantified an important ecological interaction shaping optimal organismal immune strategy. In so doing, we show that differences in magnitude and timing of reproductive output can drive differences in immune strategy, and that different longevity-associated life history traits can pull in different directions on immune strategy. But epidemiology sets the strength

and direction of the life history-immune strategy relationship for all life history traits. Thus researchers hoping to understand immune defenses and their evolution must be mindful of the interaction between overall host strategy and host environment.

## Methods

### Basic model

We use a model of immune system recognition and response strategy first developed in [8]. In this model, immune strategy is described as a trade-off between sensitivity and specificity in detection and discernment of parasite signals (Fig 1, [8]). The trade-off between sensitivity and specificity can be expressed mathematically with a receiver-operator curve (ROC), a tool frequently used in describing signal detection [8,45,46]. Sensitivity and specificity are related with the equation

$$s_e = 1 - exp(-\gamma(1 - s_p)), \tag{1}$$

where $s_e$ is immune sensitivity, $s_p$ is immune specificity, and $\gamma$ is a discrimination coefficient influenced by the overlap of the distributions of parasite and non-parasite signals and the ability of the host to discern that overlap (Fig 1B and 1C). Higher values of $\gamma$ indicate less overlap and greater discernment. Both $s_e$ and $s_p$ are constrained on the interval [0,1].

Overall mortality risk is a sum of mortality hazards from disease and non-disease causes. We include background, infection, and immunopathology mortality. Sensitivity and specificity, coupled with infection risk, influence the total mortality attributable to infection and immunopathology. Survival $s_x$ at age $x$ is expressed via the following equation:

$$s_x = exp(-[\mu_b + (1 - i_r)\mu_i(1 - s_p) + i_r\mu_d(1 - s_e) + i_r\mu_{id}s_e]), \tag{2}$$

where $i_r$ is infection risk, $\mu_b$ is background mortality risk, $\mu_i$ is the risk of immunopathology mortality when uninfected, $\mu_d$ is the risk of infection-induced mortality, assuming no immune response, and $\mu_{id}$ is the risk of infection-induced mortality when there is an immune response, including any immunopathology risk. While the various mortality parameters are constrained only to be non-negative, $i_r$ is constrained on [0,1]. Substituting Eq (1) into Eq (2) allows calculation of $s_x$ as a function of $s_p$.

When we discuss epidemiological risk in this paper, we are referring to any of $i_r$, $\mu_i$, $\mu_d$, and $\mu_{id}$. By varying these parameters with age we can explore how immune strategy affects survival at different ages when the risks differ during life. We combine the resulting survival curve with a reproductive output schedule to produce a demographic matrix with survival and fertility parameters for multiple age classes. This matrix describes the dynamics of a population with the given background mortality, epidemiological risks, reproductive output schedule, and immune strategy. Each column in the matrix represents an age class of some duration (all uniform length in our approach) and contains one parameter in the top row describing the fertility of individuals of that age class and another parameter in the subdiagonal describing their survival to the next age class. The last column has its survival parameter in the diagonal and describes all individuals in that age class and older age classes. Survival parameters for each age class $x$ are $s_x$ as calculated above. As in [44], we define the fitness of a population as $\lambda$, the dominant eigenvalue of the matrix, which is the population growth rate [51]; $\lambda$ is considered preferable to $R_0$, the net reproductive rate, in circumstances with overlapping generations like our framework [77]. By calculating $\lambda$ for a range of immune specificities and sensitivities–here $s_p$ values from 0 to 1 with intervals of 0.001 and the associated values of $s_e$–we can identify the strategy maximizing fitness. While we mostly report our results as optimal immune specificity for convenience, all values can be transformed to optimal immune sensitivities by Eq (1).

## Designating reproductive and epidemiological variation

In our exploration of the effects of reproductive demography, we defined five different schedules for reproductive output. For our baseline strategy, rate of reproduction was constant at all ages from the age of first reproduction ("Baseline"). We then chose two strategies that also featured constant rates of reproduction across age classes, but with higher ("High") or lower ("Low") yearly reproductive outputs relative to that baseline. To investigate how variation in reproduction with age might affect optimal immune strategy, we also used simple schedules where reproduction either increased with age class at a constant rate from a low initial value ("Rising") or decreased with age class at a constant rate from a high initial value ("Declining"). These reproductive schedules give the fertility parameters for each age class in the demographic matrix, beginning from the age class of first reproduction; age classes pre-reproductive maturity have a fertility of 0. For matrices built with each of these reproductive schedules we calculated the optimal immune strategy, as described above, in a particular epidemiological context where infection risk drops at the age of reproductive maturity. In addition, changing reproductive output alters fitness, such that our five schedules have different fitness values when confronted with the same mortality risks; therefore, to establish the robustness of our results, we also explored an alternate case with different background mortality rates $\mu_b$ for each schedule to approximately equalize the fitness values associated with them while still using the same epidemiological context across the five schedules.

We next considered how variation in epidemiological risk across life affects optimal immune strategy. We identified two epidemiological scenarios, each with a single change in risk at reproductive maturity–a "stepped" mode (S6 Table). For our main results we focused on variation in infection risk ($i_r$). In our scenarios $i_r$ fell (A1) and rose (A2) at reproductive maturity, respectively. For all scenarios, the other parameters remained constant. These scenarios could be held to examine a case where infection risk declines from high to low (A1) and where infection risk rises from low to high (A2). We then combined these two scenarios with the five different reproductive schedules to explore how varying epidemiological context alters the relationship between reproduction and optimal immune strategy. As previously shown, if epidemiological risk does not vary with age, features of demography do not modulate the optimal immune sensitivity and specificity in our model [44] and therefore, we do not further evaluate scenarios where epidemiological risk is flat over age.

We also explored how different magnitudes of risk, and quantities of variation in said risk, affected optimal immune specificity and sensitivity. Here we explored four different scenarios for variation in infection risk ($i_r$), again with risk changing once at reproductive maturity (S7 Table). To characterize the importance of absolute magnitude of risk, we used two scenarios where $i_r$ dropped at reproductive maturity; the amount of decline was the same between the two scenarios, but the absolute level of risk differed in each age class, with a low-risk scenario (B1) and a high-risk scenario (B2). To explore how the magnitude of risk variation affected optimal strategy, we used two scenarios with different quantities of risk change at reproductive maturity, with a large-change scenario (B3) and a small-change scenario (B4). For all scenarios, the other parameter values in our model were $\mu_b = 0.15$, $\mu_d = 0.3$, $\mu_i = 0.1$, $\mu_{id} = 0.01$, and $\gamma = 4$. As above, we combined these scenarios with the five reproductive schedules to investigate how magnitude and variation of risk affect optimal immune strategies.

To further consider the importance of which risk parameters vary with age and how they do so, we also considered two alternative types of variation in risk with age. For the first type, we varied undetected infection mortality risk ($\mu_d$) with age while infection risk ($i_r$) was held constant. We repeated each of the A and B scenarios for $i_r$ with variation in $\mu_d$ instead. In these scenarios, $i_r = 0.4$ at all ages, while $\mu_d$ varies on the same intervals that $i_r$ did in the analogous

scenarios, and all other parameters retain the same values. For the second type, we defined a separate mode of variation–"smoothed"–in which the increase or decrease in risk between age classes is constant. To translate between our scenarios, we used the risk value prior to reproduction in our stepped scenarios as the risk value for the first age class in our smoothed scenarios, and we used the risk value for reproductive years in the stepped scenarios as the risk value for the last age class in our smoothed scenarios. We used this approach with both $i_r$ and $\mu_d$, with the same values for other parameters. We therefore explored versions of A1–A2 and B1–B4 for both the stepped and smoothed modes and for both $i_r$ and $\mu_d$.

## Predicting optimal immune defense from population projection matrices

To root our analysis in existing empirical schedules of reproduction and survival, we leveraged a large representative dataset of population projection matrices from the COMADRE database [52,53] to serve as starting points for the analysis. These matrices give us survival and fertility parameters calculated from wild populations of various animal species. We combined these parameters, with some transformation, with epidemiological risk parameters and then explored a range of specificity ($s_p$) and sensitivity ($s_e$) values–again from 0 to 1 with intervals of 0.001 –to determine what immune specificity and sensitivity would produce the maximum population growth rate given those risk, fertility, and survival parameters.

We restricted our analysis to primitive, irreducible, and ergodic COMADRE matrices (i.e., the matrix diagonal contains 0s except for the final entry, survival is contained in the off-diagonals, and fertility is contained in the first row) that are structured by age in years and for which each age class is equivalent to one year. By using only matrices structured in this manner we ensured comparability between populations in epidemiology and life history. After this filtering, we were left with 298 population matrices representing 129 chordate species. We also report results for an analysis that includes only 151 population matrices representing 47 mammal species.

## Transforming matrices for use with prediction method

The population projection matrices from COMADRE can be of varying dimension, depending on both the demography of the population and the data available. For matrices with each age class equivalent to one year, which we are using, this primarily affects the amount of detail in describing the population dynamics of later age classes. Because we are interested in how epidemiological risk variation across life affects immune strategy and accordingly want to ensure that the epidemiological schedules are comparable across taxa, we standardized matrix dimension with respect to checkpoints in the life history of an organism. We transformed each original matrix to a new one with its transformed dimension being the sum of the age at first reproduction of the organism and the reproductive life expectancy of the organism. In this way we can scale epidemiological risk variation over age to each species's lifespan while retaining comparability across taxa with respect to life history.

Re-sizing matrices according to our above criterion requires increasing the dimension of some matrices and decreasing the dimension of others. Increasing dimension is simple: we added columns and rows with the same survival and fertility parameters as those in the final column and row of the original matrix. Such a matrix describes the exact same population dynamics, including population growth rate (λ), as the original matrix, because the final age class of any matrix describes the survival and fertility of any individual of that age class or older. Decreasing dimension is a slightly more complex process. To accomplish this, we used an algorithm designed by Hooley [78]. In brief, this algorithm collapses designated rows and columns by taking an average of the survival and fertility parameters in those rows and

columns and weighting by stable age structure in those age classes. The weighted average parameters are used as the fertility and survival parameters for a single new column replacing the collapsed columns. We follow the recommendation from Salguero-Gómez and Plotkin [79] to collapse the rows and columns that represent the oldest age classes, as this criterion minimizes the distortion of demographic patterns by only affecting the degree of nuance in later age classes, which will generally have relatively small contributions to population dynamics. This collapsing algorithm produces a matrix of the desired dimension with population dynamics quite tightly approximating those of the original matrix. Overall 288 of the 298 matrices in our dataset required resizing, but this resizing has minimal impact on estimates of population dynamics for each matrix (S3 Fig).

While the COMADRE matrices do provide the age trajectory of survival (and thus cumulative mortality hazard) for our focal species, they offer no insight into causes of mortality. In theory one would want to determine immune strategy based on a precise knowledge of infection and immunopathology-associated risk. But there is no way to establish what fraction of the mortality recorded in the matrix is attributable to infectious disease or immunopathology vs. other causes of death. To address this, we must define our own background mortality and epidemiological risk parameters, and these can only be loose approximations of the risks experienced by a given organism as recorded in the relevant population projection matrix. Because it has previously been established that epidemiological context strongly affects immune strategy in this model framework [8,44], we want to compare among species subject to the same epidemiological risks to understand how life history might be linked to immune specificity and sensitivity without confounding. We therefore choose arbitrary epidemiological risk parameters (and schedules of variation thereof) which can be defined consistently for all matrices, although in natural systems we would expect variation in risks.

Our remaining problem is choosing background mortality parameter values. Because we are deriving life history from the original population projection matrix, we want to choose our value for background mortality ($\mu_b$) based on the mortality recorded in the original matrix. We therefore designate $\mu_b$ for each age class of our new matrix as being some proportion $\rho$ of the original total mortality for that age class as given in the original matrix. This designation arbitrarily defines $\rho$ as the proportion of overall mortality attributable to background causes. Thus the $\mu_b$ value for each given age class $x$ in our new survival curve is calculated as a product of the logarithm of survival $s_x$ in the relevant age class $x$ of the re-sized matrix and a constant $\rho$, such that

$$\mu_b = -\rho(log(s_x^{orig})). \tag{3}$$

For our analysis we used $\rho = 0.75$, but the value of $\rho$ has a minimal effect on our results. We combine the new $\mu_b$ value for each age class calculated as in Eq (3) with epidemiological risks to produce a new survival curve for the population under consideration. The resulting survival curve only approximates the original survival curve, but it allows us to investigate different epidemiological scenarios and immune strategies.

Because the qualitative results from variation in infection risk ($i_r$) and undetected infection mortality risk ($\mu_d$) are the same in our earlier analyses, we only looked at different $i_r$ scenarios for this portion of our analysis. We considered two ways that $i_r$ might vary with age. The first is identical to our stepped mode described earlier: we define one $i_r$ value for pre-reproductive age classes and another $i_r$ value for reproductive age classes. Our second mode is similar to our smoothed mode previously described in allowing $i_r$ to change smoothly across lifespan. Here, we define a starting risk for the schedule, which would be $i_r$ in the first age class of the matrix, and an ending risk for the schedule, which would be $i_r$ for the last age class of the matrix, and

allow a uniform change in $i_r$ across intermediate age classes. However, since each matrix differs in dimension (and therefore absolute time and age described), this rate of change in $i_r$ cannot be the same from matrix to matrix. Therefore we scale the rate of change by the dimension of the matrix, such that change in $i_r$ relative to the demography described by the matrix–demography that is standardized as the sum of age at first reproduction and reproductive life expectancy–is constant across all populations (S4 Fig). An example of how this works: in a matrix with dimension 3 (i.e. having 3 age classes), a schedule for $i_r$ might be [0.45, 0.325, 0.2], while for a matrix of dimension 6 the equivalent schedule for $i_r$ would be [0.45, 0.4, 0.35, 0.3, 0.25, 0.2]. As noted above, the absolute pace of change of risk varies between the two schedules, but the relative pace of change with respect to life history is constant and the absolute change across life is also constant (S4 Fig).

In this way we produce equations for survival $s_x$ for each age class in the matrix. With these, and the fertility parameters of the original matrix, we determined optimal specificity as described above, creating matrices across the range of values of $s_p$ and identifying the values of $s_p$ and $s_e$ that produce $\lambda_{max}$. This is our predicted optimal immune specificity and sensitivity associated with the original matrix from the COMADRE database. As above, we present our results in terms of specificity, but $s_p$ values can be easily translated to $s_e$ values by Eq 1.

## Linear model for relationship between demography and predicted immune strategy

To describe the relationship between life history and immune specificity, we calculated four life history summary statistics from our matrices: age class of first reproduction, life expectancy post-reproductive maturity or reproductive life expectancy, mean reproductive rate, and generation time. Each of these statistics is calculated in the manner defined by [27]. To describe the relationship between the life history traits and the predicted optimal immune specificity, we used a Bayesian linear model of specificity as a function of life history trait values. For our main results we used life history statistics as calculated from the original matrix in the COMADRE database, rather than the transformed, optimized matrix, to counteract any systematic distortions introduced by our manipulation of matrix dimension and survival curve calculation (although note as above that dimension manipulation does not seem to induce any demographic distortions). However, to ensure robustness we also did repeat our analysis using statistics calculated from the post-processing matrices. In addition, we considered generation time separately from the other three traits because in our dataset it is highly collinear with each of them.

Life history trait parameter values were first log-transformed to produce trait value distributions approximating a normal distribution. They were then standardized by calculating Z-scores from the mean and standard deviation of the sampled values for each parameter. Predicted optimal immune sensitivity $s_p^*$ was left unstandardized because it has no units and is constrained on the interval [0,1]. We also assumed for our linear model that $s_p^*$ is normally distributed; while $s_p$ does have a constrained possible range, our model predictions never approached those boundaries, such that a normal distribution can reasonably be used. The model was constructed with the "ulam" function of the "rethinking" package v2.01 (which implements RStan) in R v4.0.1 and run for 2000 samples [80,81]. We designated the prior for the intercept of $s_p^*$ as a normal distribution with $\mu = 0.5$ and $\sigma = 0.15$; for all coefficients, we similarly used a normal distribution prior centered on $\mu = 0$, with $\sigma = 0.1$. In total we examined 121 different scenarios for variation in infection risk, with differences in how much infection risk changed across life and whether it rose or fell. We produced predicted optimal immune specificities for our set of population projection matrices, and we analyzed the results with

linear models for each scenario. We repeated these analyses for both stepped and smoothed modes of $i_r$ variation.

## Supporting information

**S1 Fig. Demographic details for reproductive schedules with optimal immune strategies.** Optimal immune strategies identified in Fig 2 and S1 Table. Dashed line shows age class of reproductive maturity, the third age class. A) Reproductive value distributions, with reproductive value in the first age class defined as 1; B) Stable age structures; C) Elasticities of λ with respect to survival; D) Elasticities of λ with respect to fertility. Infection risk $i_r$ drops from 0.6 in pre-reproductive age classes to 0.2 in reproductive age classes. Other parameter values are $\mu_b = 0.15$, $\mu_i = 0.1$, $\mu_d = 0.3$, $\mu_{di} = 0.01$, and γ = 4.
(PDF)

**S2 Fig. Epidemiological environment alters the effect of reproduction on immune strategy: infection mortality risk $\mu_d$.** Reproduction begins in the third age class for all schedules. In each scenario, $\mu_d$ starts at one value and drops to a lower value at the third age class. A) The change in optimal immune specificity associated with differences in epidemiological context (i.e. changes in $\mu_d$, on the x-axis) and reproduction (different points and lines, color-coded at center). Parameter values are $\mu_b = 0.15$, $\mu_i = 0.1$, $\mu_{id} = 0.01$, $i_r = 0.4$, and γ = 4. In the declining scenario, $\mu_d$ drops with age from 0.6 to 0.2; in the rising scenario, $\mu_d$ increases from 0.2 to 0.6. B) The change in range of optimal specificities associated with different reproductive demographies associated with different magnitudes of variation in decline of infection mortality risk $\mu_d$ with age. Parameter values are $\mu_b = 0.15$, $\mu_i = 0.1$, $\mu_{id} = 0.01$, $i_r = 0.4$, and γ = 4. In the lower range scenario, $\mu_d$ declines from 0.45 to 0.2; in the higher range, from 0.7 to 0.45; in the broad range, from 0.7 to 0.2; in the narrow range, from 0.525 to 0.375.
(PDF)

**S3 Fig. Implementation of matrix manipulation methods does not distort demographic patterns.** A) Histogram shows, for the 298 matrices used from the COMADRE database, how they were altered in dimension, either expanded or collapsed, and the change in dimension after alteration. B) Plot shows the change in matrix dimension and the associated difference between λ when calculated for the original matrix and when calculated for the matrix after the dimension has been altered.
(PDF)

**S4 Fig. Method for scaling risk parameter change to matrix dimension.** X-coordinates for lines in absolute age plots determined by age classes within matrix, and so the blue line is for an organism with a short lifespan and smaller matrix dimension, while the red line is for an organism with a long lifespan and larger matrix dimension. Dimension of manipulated matrices (see S3 Fig) determined as sum of age at first reproduction and reproductive life expectancy.
(PDF)

**S5 Fig. Influence of reproductive demography on optimal immune strategy: smoothed infection risk $i_r$ variation.** Plot showing optimal combination of immune sensitivity and specificity for each of five reproductive demographic schedules (color-coded at left), for a single epidemiological risk environment where infection risk declines at a constant rate from $i_r = 0.6$ in age class 1 to $i_r = 0.2$ in age class 10. Reproduction begins in the third age class for all schedules. Strategy optima, shown as points on the dashed curve, are determined as the immune specificity and sensitivity maximizing λ, the population growth rate. Dashed curve shows the shape of the specificity/sensitivity trade-off curve for γ = 4. Solid lines show values of the

respective optimal strategies on each axis. Other parameter values are $\mu_b = 0.15$, $\mu_i = 0.1$, $\mu_d = 0.3$, and $\mu_{id} = 0.01$.
(PDF)

**S6 Fig. Epidemiological environment alters the effect of reproduction on immune strategy: infection risk $i_r$ with smoothed risk variation.** Reproduction begins in the third age class for all schedules, and $i_r$ changes a constant amount from age class to age class within each schedule. A) The change in optimal immune specificity associated with differences in epidemiological context (i.e. changes in $i_r$, on the x-axis) and reproduction (different points and lines, color-coded at center). Parameter values are $\mu_b = 0.15$, $\mu_i = 0.1$, $\mu_d = 0.3$, $\mu_{id} = 0.01$, and $\gamma = 4$. In the declining scenario, $i_r$ declines with age from 0.6 to 0.2; in the rising scenario, $i_r$ increases from 0.2 to 0.6. B) The change in range of optimal specificities associated with different reproductive demographies associated with different magnitudes of variation in decline of infection risk $i_r$ with age. Parameter values are $\mu_b = 0.15$, $\mu_i = 0.1$, $\mu_d = 0.3$, $\mu_{di} = 0.01$, and $\gamma = 4$. In the lower range scenario, $i_r$ declines with age from 0.45 to 0.2; in the higher range, from 0.7 to 0.45; in the broad range, from 0.7 to 0.2; in the narrow range, from 0.525 to 0.375.
(PDF)

**S7 Fig. Epidemiological environment alters the effect of reproduction on immune strategy: infection mortality risk $\mu_d$ with smoothed risk variation.** Reproduction begins in the third age class for all schedules, and $\mu_d$ changes a constant amount from age class to age class within each schedule. A) The change in optimal immune specificity associated with differences in epidemiological context (i.e. changes in $\mu_d$, on the x-axis) and reproduction (different points and lines, color-coded at center). Parameter values are $\mu_b = 0.15$, $\mu_i = 0.1$, $\mu_{id} = 0.01$, $i_r = 0.4$, and $\gamma = 4$. In the declining scenario, $\mu_d$ declines with age from 0.6 to 0.2; in the rising scenario, $\mu_d$ increases from 0.2 to 0.6. B) The change in range of optimal specificities associated with different reproductive demographies associated with different magnitudes of variation in decline of infection mortality risk $\mu_d$ with age. Parameter values are $\mu_b = 0.15$, $\mu_i = 0.1$, $\mu_{id} = 0.01$, $i_r = 0.4$, and $\gamma = 4$. In the lower range scenario, $\mu_d$ declines with age from 0.45 to 0.2; in the higher range, from 0.7 to 0.45; in the broad range, from 0.7 to 0.2; in the narrow range, from 0.525 to 0.375.
(PDF)

**S8 Fig. Interaction of demography and epidemiology: smoothed infection risk $i_r$ variation.** Infection risk is set such that infection risk changes at a constant rate relative to lifespan from a defined value of $i_r$ for the first age class to a defined value of $i_r$ for the last age class. Our dataset comprises 298 population matrices representing 129 chordate species. For all scenarios, parameter values are $\mu_d = 0.3$, $\mu_i = 0.1$, $\mu_{id} = 0.01$, $\rho = 0.75$, and $\gamma = 4$. A) Predicted optimal immune specificities when infection risk declines with age, with respect to population reproductive life expectancy and mean reproductive rate as calculated from original matrix. Infection risk $i_r$ prior to reproductive maturity is 0.45; for reproductive age classes, it is 0.2. B) Predicted optimal immune specificities when infection risk rises with age with respect to population reproductive life expectancy and mean reproductive rate as calculated from original matrix. Infection risk $i_r$ prior to reproductive maturity is 0.2; for reproductive age classes, it is 0.45.
(PDF)

**S9 Fig. Epidemiological dependency of life history-specificity relationship: smoothed infection risk $i_r$ variation.** Tile plots showing, for a variety of scenarios of variation in $i_r$, coefficients of relationship between the designated life history trait and predicted optimal immune specificity as estimated from Bayesian linear models. Each tile is a scenario in which predicted optimal immune specificities were generated for 298 population projection matrices

representing 129 chordate species. A linear model was used to estimate relationship coefficients for each scenario. The coefficient value in the plot represents the mean value of the posterior probability distribution, except on the diagonal; on the diagonal, all coefficients are 0 (see text). The value of $i_r$ on the x-axis is the risk for the first age class in the matrix, while the value on the y-axis is the risk for the last age class; rate of change between intermediate age classes is adjusted per matrix, based on dimension, so that risk changes at a constant rate from age class to age class within a matrix but the absolute magnitude of risk change is equivalent for all matrices. Life history trait values log-transformed and standardized as Z-scores for comparability of coefficients. For all estimated coefficients off the diagonal, 89% credible intervals do not include 0. For all scenarios, parameter values are $\mu_d = 0.3$, $\mu_i = 0.1$, $\mu_{id} = 0.01$, $\rho = 0.75$, and $\gamma = 4$. Unlike Fig 5, age class of first reproduction is not shown because our models do not confidently predict a relationship with immune specificity for any $i_r$ scenario.
(PDF)

**S1 Table. Effect of background mortality $\mu_b$ on optimal immune strategy.** Comparison of results from analysis of effect of reproductive schedule on optimal immune strategy when $\mu_b$ does not change to equalize $\lambda$ (original) and when $\mu_b$ does change to equalize $\lambda$ (adjusted). $s_p^*$ is the optimal immune specificity that maximizes $\lambda$. Infection risk $i_r$ declines from 0.6 before reproductive maturity (age classes 1 and 2) to 0.2 after (classes 3+). Other parameter values are $\mu_i = 0.1$, $\mu_d = 0.3$, $\mu_{id} = 0.01$, and $\gamma = 4$.
(DOCX)

**S2 Table. Effect of background mortality $\mu_b$ on optimal immune strategy: smoothed infection risk variation.** Comparison of results from analysis of reproductive schedule on optimal immune strategy when $\mu_b$ does not change to equalize $\lambda$ (original) and when $\mu_b$ does change to equalize $\lambda$ (adjusted). $s_p^*$ is the optimal immune specificity that maximizes $\lambda$. Infection risk $i_r$ declines smoothly from 0.6 in the first age class to 0.2 in the final age class. Other parameter values are $\mu_i = 0.1$, $\mu_d = 0.3$, $\mu_{id} = 0.01$, and $\gamma = 4$.
(DOCX)

**S3 Table. Results from Bayesian linear model for demography and immune strategy–analysis using only mammal life histories.** Linear model looks at model-predicted optimal immune specificity as a function of three different life history summary statistics. Results are means and, in brackets, boundaries of 89% highest posterior density intervals (HPDI) for posterior probability distributions for parameter values. Entries in *italics* indicate the 89% HPDI overlaps with 0 for that parameter. All summary statistics were calculated from original matrix in COMADRE database, log-transformed, and standardized as Z-scores. The only matrices included are those 151 qualifying matrices from 47 different mammal species. For stepped epidemiological scenario, when infection risk ($i_r$) is rising, $i_r$ in pre-reproductive years is 0.2, and $i_r$ in reproductive years is 0.45. When infection risk declines in the stepped scenario, $i_r$ in pre-reproductive years is 0.45, and $i_r$ in reproductive years is 0.2. In smoothed declining scenario, $i_r$ declines from 0.45 to 0.2; in rising scenario, $i_r$ rises from 0.2 to 0.45. Other parameter values are $\mu_d = 0.3$, $\mu_i = 0.1$, $\mu_{id} = 0.01$, $\rho = 0.75$, and $\gamma = 4$.
(DOCX)

**S4 Table. Results from Bayesian linear model for demography and immune strategy–analysis with generation time only.** Linear model looks at model-predicted optimal immune specificity as a function of the common life history summary statistic generation time, which is heavily confounded with other summary statistics. Results are means and, in brackets, boundaries of 89% highest posterior density intervals. All summary statistics were calculated from original matrix in COMADRE database, log-transformed, and standardized as Z-scores.

Dataset includes 298 qualifying matrices from 129 chordate species. In smoothed declining scenario, infection risk ($i_r$) declines from 0.45 to 0.2; in rising scenario, $i_r$ rises from 0.2 to 0.45. For stepped epidemiological scenario, when infection risk is rising, $i_r$ in pre-reproductive years is 0.2, and $i_r$ in reproductive years is 0.45. When infection risk declines in the stepped scenario, $i_r$ in pre-reproductive years is 0.45, and $i_r$ in reproductive years is 0.2. In smoothed declining scenario, $i_r$ declines from 0.45 to 0.2; in rising scenario, $i_r$ rises from 0.2 to 0.45. Other parameter values are $\mu_d = 0.3$, $\mu_i = 0.1$, $\mu_{id} = 0.01$, $\rho = 0.75$, and $\gamma = 4$.
(DOCX)

**S5 Table. Results from Bayesian linear model for demography and immune strategy–analysis with post-processing life history statistics.** Linear model looks at model-predicted optimal immune specificity as a function of three different life history summary statistics. Results are means and, in brackets, boundaries of 89% highest posterior density intervals (HPDI) for posterior probability distributions for parameter values. Entries in *italics* indicate the 89% HPDI overlaps with 0 for that parameter. All summary statistics were calculated from output matrices after manipulation and optimization of $s_p$ (unlike results shown in Fig 4 and Table 1), log-transformed, and standardized as Z-scores. Dataset includes 298 qualifying matrices from 129 chordate species. For stepped epidemiological scenario, when infection risk ($i_r$) is declining, $i_r$ in pre-reproductive years is 0.2, and $i_r$ in reproductive years is 0.45. When infection risk declines in the stepped scenario, $i_r$ in pre-reproductive years is 0.45, and $i_r$ in reproductive years is 0.2. In smoothed declining scenario, $i_r$ declines from 0.45 to 0.2; in rising scenario, $i_r$ rises from 0.2 to 0.45. Other parameter values are $\mu_d = 0.3$, $\mu_i = 0.1$, $\mu_{id} = 0.01$, $\rho = 0.75$, and $\gamma = 4$.
(DOCX)

**S6 Table. Epidemiological Risk Scenarios: Set A.** The parameter values for each different analysis of the reproductive demography-immune strategy relationship as described in the Methods. We also considered similar scenarios in which infection mortality risk ($\mu_d$) varies on the same intervals as infection risk ($i_r$) does here, while $i_r$ is held constant at 0.4. For Figs 3 and S2, risk parameters changed from the first value in brackets for pre-reproductive age classes (classes 1 and 2) to the second for reproductive age classes (classes 3+). For S5 and S6 Figs, infection risk smoothly changed from the first value in the brackets in the first age class to the second value in the last age class of the matrix, with the same change in risk between each age class.
(DOCX)

**S7 Table. Epidemiological Risk Scenarios: Set B.** The parameter values for each scenario of the reproductive demography-immune strategy relationship as described in the Methods, used for an analysis of the influence of magnitude of change in risk with age on immune strategy. We also considered scenarios where infection mortality risk ($\mu_d$) varied on the same intervals as infection risk ($i_r$) does here, with $i_r$ then being held at 0.4 across all age classes. For Figs 3 and S2, risk parameters dropped from the first value in brackets for pre-reproductive age classes (classes 1 and 2) to the second for reproductive age classes (classes 3+). For S5 and S6 Figs, infection risk smoothly fell from the first value in the brackets in the first age class to the second value in the last age class of the matrix, with the same change in risk between each age class.
(DOCX)

**S1 Code. R code necessary for running simulations to find optimal immune specificity and sensitivity and recreating figures and tables in the main text and supplement.**
(R)

## Acknowledgments

We thank C. Riehl, B. vonHoldt, and the Graham Lab for valuable discussions.

## Author Contributions

**Conceptualization:** Alexander E. Downie, Andreas Mayer, C. Jessica E. Metcalf, Andrea L. Graham.

**Investigation:** Alexander E. Downie, Andreas Mayer, Andrea L. Graham.

**Methodology:** Alexander E. Downie, C. Jessica E. Metcalf, Andrea L. Graham.

**Supervision:** C. Jessica E. Metcalf, Andrea L. Graham.

**Writing – original draft:** Alexander E. Downie.

**Writing – review & editing:** Alexander E. Downie, Andreas Mayer, C. Jessica E. Metcalf, Andrea L. Graham.

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
