## [Decision Letter · Decision Letter 0]

25 Oct 2021

Dear Mr. Downie,

Thank you very much for submitting your manuscript "Optimal immune specificity at the intersection of host life history and parasite epidemiology." for consideration at PLOS Computational Biology. As with all papers reviewed by the journal, your manuscript was reviewed by members of the editorial board and by several independent reviewers. The reviewers appreciated the attention to an important topic. Based on the reviews, we are likely to accept this manuscript for publication, providing that you modify the manuscript according to the review recommendations.

Sincerely,

Miles P. Davenport, MB BS, D.Phil

Associate Editor

PLOS Computational Biology

Nina Fefferman

Deputy Editor

PLOS Computational Biology

[LINK]

Reviewer's Responses to Questions

**Comments to the Authors:**

Reviewer #1: In the manuscript “Optimal immune specificity at the intersection of host life history and parasite epidemiology”, the authors applied their previously-developed model to investigate the optimal immune strategies in terms of immune specificity by varying the reproduction schedules and key parameters such as the infection risk i_r. Further incorporating data from COMADRE Animal Matrix Database, the authors explored the association between the optimal immune strategies and three life history traits, suggesting that both demographic and epidemiological factors can greatly alter the optimal immune strategies. Overall, this work is solid and provides some interesting and useful insights into how the immune strategies change against epidemiological and demographic variations. On the other hand, I feel that the presentation could be improved to better explain how the computational model is implemented – the high-level description of matrix computation is not very clear, as it is very important in the modelling part. While most of the significant results depends on the previously-developed model and the associated model assumptions, the significance of computational work needs to be explicitly emphasized. Also the Discussion section is unnecessarily long and therefore loses its focus. Therefore, I suggest the authors revising the manuscript so that readers can better link the results to the model set-up, and therefore better understand the interpretation of the results.

Below are some further comments:

1. Figure 1 (C) I looks like a histogram instead of a curve to me. And it is not very clear how (B) and (C) are connected – how the immune response threshold in (C) is reflected in (B)?

2. Line 179 – What is “elasticity of lambda”? How is it defined? This quantity also appears in Figure S3 (C) and (D) and may need an explicit definition.

3. Age class is intensely discussed in the Results section is the key forming matrices for computation – what do they represent in different subsections? In all the cases studied, do they have the same structure of age classes? When incorporating the data, it seems that the age class is on a yearly basis, is it the same for the other computational cases?

4. The reproductive schedule is one of the key aspects in the model. However, how different reproductive schedules are implemented in the computational model (assuming it’s in the Leslie matrix?) is not well explained, details of this in the Methods section will be helpful for readers to understand the modelling system.

5. Line 212 – “i_r declines drops” keep either “declines” or “drops”

6. Figure 3A) Why the baseline value of i_r is different in the declining and rising cases? The message this figure conveys is the association between the change of infection risk and the optimal immune specificity under different reproductive schedules, it may be more convincing if these comparisons start with a same baseline value. The same concern applies to the results in Figure 4 as well.

7. Line 230 – “The epidemiology is essential” This sentence is redundant – the point has been well made in the previous sentence.

8. Results for Figure 4 are based on fixed parameter values of mu_d, mu_i, mu_id, pho, and gamma across 129 chordate species, which implicitly assumes that the immune responses for these species are the same. Is this assumption supported by any study? Because the optimal immune strategies could also be sensitive to these parameters. Why would you like to choose these particular values?

9. Line 350 –“while when” – either will be fine

10. Line 431 – “different their immune specificities” – “different immune specificities”

11. Overall the Discussion section is too long from my point of view. For example, paragraph discussing challenges and further benefits contain too many details. In particular, there are three “for example” in the paragraph discussing further benefits. These paragraphs can be shortened and combined.

12. I find Figure S2 is a bit confusing. Are you trying to demonstrate the scaling of i_r, which is done by interpolating start and end points?

Reviewer #2: This study is a useful extension of prior models to investigate the evolution of immune specificity, assuming a tradeoff between sensitivity and specificity, i.e., pathology from infection or immunity. Several of the investigators had previously shown how age at reproduction could affect the optimal strategy. Here, the investigators examine how life-stage-varying infection risk interacts with life history to influence optimal specificity. Using a simple mathematical model, they show that changes in infection risk over a typical lifespan can completely flip the ranking of strategies (Fig 3). They then use empirical matrix population models from the COMADRE database to evaluate, given uncertainty in actual infection risk over time in different species, how the optimal specificity correlates with the mean reproductive rate and other demographic traits. They show again that life history alone is a poor predictor of specificity (Fig 4).

This is a careful analysis that should interest infectious disease, population, and evolutionary biologists and some evolutionarily minded immunologists. Host immune systems are under strong selective pressures, and yet there is little integrative theory that attempts to explain the diversity of strategies that are observed (or that probably would be observed, if we studied other species as well as we should). Although I like to think we will have a more nuanced understanding of actual immunological tradeoffs in a few decades, this is a logical start. The most important outcomes of this paper could be increased attention not only to potential tradeoffs, including immunopathology in other species, but also to the quantitative impact of pathogens on survival in different life stages.

I have only minor suggestions for the paper, which presents a solid contribution in its current form. My primary suggestions are of the "but what about..." variety: What if pathogens' fitness impacts and/or immunopathology risk were mostly through fecundity rather than survival? How much could gamma and optimal specificity vary by life stage? Do infection risks with multigenerational or inconsistent periods (truly time-varying) change the optimal strategy? How does coevolution affect the optimal strategy? I am most curious about the first, although the Discussion is long enough, and life-stage-varying sensitivity is briefly mentioned.

My second general suggestion is that the authors consider providing more examples to support the existence of a tradeoff and potentially some background on population matrices. Few readers will be familiar with both areas.

My remaining suggestions involve possible clarifications and typos:

The authors may want to consider how Schnaack & Nourmohammad's results (eLife, 2021; https://elifesciences.org/articles/61346v1), which are effectively about sensitivity from immune memory, fit in with the conclusions.

ll. 385-399: R0 is often a better indicator than lambda of long-term competitive success, right? This paragraph suggests it is not important, but I find the passage confusing.

Figure 1: Panels B and C are reversed between the figure and caption.

Figure 2: I found the horizontal and vertical lines in the right panel confusing. (I know a few other readers who felt the same way.) Why not just show points?

"Robusticity", not "robustness"?

Reviewer #3: This well written paper addresses theoretically the interesting question of what determines optimal immune investment and moreover why there is so much variation amongst hosts both within and between species. It makes an important contribution to this field examining in particular the relative investment in sensitivity and specificity of the immune system. It follows up on this framing of the problem by some of the previous authors in their Nature Comms paper and I think the characterization of sensitivity – how likely to fire the immune response – and specificity – how tightly aligned to the pathogen the response is – is very interesting to explore. Here they propose age structured models that include epidemiology – in the sense that they look at disease risk although they don’t have epidemiological feedbacks with a dynamical disease modeling approach – to examine how life-history and epi interact to determine variation in these traits. The use of the COMADRE demographic data set is very novel and important and they have a number of important results that help explain patterns in nature and will drive future research. The results in figure 3 are really interesting. This is exactly what you want theory like this to do. The usefulness of the approach is emphasized by the nice example they quote in the abstract of whether infection risk increases or decreases at maturity changes the sign of the selection on immune specificity. I think it is a good contribution and should be published.

I have some detailed comments throughout that I outline in order below. Overall one criticism that I have is that it is not always clear that the paper is looking at a particular aspect of the immune response. One interesting about the paper is that it is looking at sensitivity and specificity, but at times immune response is used in very general terms. I think this can be a bit confusing for the reader as a lot of other work is focused on some sort of absolute amount of resistance or tolerance to infection. Also some readers may not equate immune response to sensitivity and specificity that you are addressing here. I understand why the authors want to be general, but I don’t think there is the need for such general language here. The sensitivity/specificity idea is central and very novel in itself and I think being clear about this would give the paper a bigger impact. It’s arguably more realistic and useful to be thinking about sensitivity/specificity rather than some absolute amount of immune response, so making this point clearly would be really useful I think. My second overall comment is that it could be made clearer that epidemiology here is not looking at epidemiological feedbacks on evolution due to an underlying dynamical disease model. This is explicitly stated in the middle of the paper, but I think it would be useful to make this point earlier, particularly given the comparison to previous models – many of which have these feedbacks although some like Shudo et al and Hamilton et al don’t. I don’t see this as a problem, but I think it would make the comparison to previous work clearer.

Abstract

I didn’t find it that compelling that you could predict a priori zoonotic risky hosts or those at risk – you make some interesting arguments based on ideas of relative sensitivity and specificity of humans and zoonotic hosts, but I wouldn’t have these statements in the abstract. For me the paper makes a big contribution to our understanding of patterns seen in nature and the fact that you get these changes in the sign of selection is really important. I think your abstract should reflect those results.

Introduction

Line 48 – consider something different to “vast” share..

Line 55 – There is a lot of different terminology in the field and I was wondering why you used facultative rather than induced here. Are you trying to say different things or is it more being careful with language that is used in immunology in specific ways?

Line 55 – it might be useful to start this sentence with “given immunopathology” or something as a reminder – some readers may not find this intuitive

Line 65 - this is a good description of previous work – but it may be good here to contrast explicitly eco(epi)- evo models with dynamical underlying disease models that examine ecological feedbacks and those that don’t – 9&10 don’t have the feedbacks – I think spending some more time clarifying that here would be useful to place this work in the field.

Line 95 – I think it is interesting to talk about herd immunity here, but more generalpoint is that prevalence is changed in these dynamical models even without acquired immunity

Line 106 – the Cape Buffalo has specific data that make it stand out, but this general point of age related impacts could be more generally referenced

Line 131 – you use maximization of population growth rate as your fitness measure where as much o f the field is using ESS type resistance to invasion type analyses – I think it is important to explain this – you don’t have eco-evo feedbacks so it makes sense but it is worth explaining .

Line 136 – Background mortality can impact optimal immune defence even if risks don’t vary with age in other modeling frameworks – as you describe elsewhere – you should make it clear that this just applies to your approach.

Line 276 – I would consider not using the acronyms MRR, AFR and RLE – they are pretty informative spelled out and not used that much. Most readers are not going to know them.

Line 346 – using mutable may confuse some readers here

Line 385. – Donnelly is looking at a different definition of resistance – I think it would be useful to discuss this.

Line 388 – Useful to explain ‘elasticity’ here.

**Have the authors made all data and (if applicable) computational code underlying the findings in their manuscript fully available?**

Reviewer #1: Yes

Reviewer #2: Yes

Reviewer #3: Yes

PLOS authors have the option to publish the peer review history of their article (what does this mean?). If published, this will include your full peer review and any attached files.

Reviewer #1: **Yes: **Wang Jin

Reviewer #2: No

Reviewer #3: No

Figure Files:

Data Requirements:

Reproducibility:

References:

---

## [Editor Report · Decision Letter 1]

2 Dec 2021

Dear Mr. Downie,

We are pleased to inform you that your manuscript 'Optimal immune specificity at the intersection of host life history and parasite epidemiology.' has been provisionally accepted for publication in PLOS Computational Biology.

(note from Ed: If you could give a written description of parameters each time they are used (especially in discussion), with the Greek letters in parentheses (rather than just Greek letters assuming the reader remembers what they are) - this would make the discussion clearer)

Best regards,

Miles P. Davenport, MB BS, D.Phil

Associate Editor

PLOS Computational Biology

Nina Fefferman

Deputy Editor

PLOS Computational Biology

---

## [Editor Report · Acceptance letter]

16 Dec 2021

PCOMPBIOL-D-21-01371R1 

Optimal immune specificity at the intersection of host life history and parasite epidemiology.

Dear Dr Downie,

I am pleased to inform you that your manuscript has been formally accepted for publication in PLOS Computational Biology. Your manuscript is now with our production department and you will be notified of the publication date in due course.

With kind regards,

Livia Horvath
